# A Method and Formula for the Quantitative Analysis of the Total Bioactivity of Natural Products

**DOI:** 10.3390/ijms24076850

**Published:** 2023-04-06

**Authors:** Shintu Mathew, Ritesh Raju, Xian Zhou, Francis Bodkin, Suresh Govindaraghavan, Gerald Münch

**Affiliations:** 1Pharmacology Unit, School of Medicine, Western Sydney University, Campbelltown, NSW 2560, Australia; 2NICM Health Research Institute, Western Sydney University, Westmead, NSW 2145, Australia; 3IMCD Australia Pty Ltd., Orchard Hills, NSW 2748, Australia

**Keywords:** herbal medicines, natural products, purification, extracts, inflammation, bioactivity, quantitative analysis, medicinal plants, chromatography, total bioactivity

## Abstract

Identification of bioactive natural products from plants starts with the screening of extracts for a desired bioactivity such as antimicrobial, antifungal, anti-cancer, anti-inflammatory, or neuroprotective. When the bioactivity shows sufficient potency, the plant material is subjected to bio-activity-guided fractionation, which involves, e.g., sequential extraction followed by chromatographic separation, including HPLC. The bioactive compounds are then structurally identified by high-resolution mass spectrometry and nuclear magnetic resonance (NMR). One of the questions that come up during the purification process is how much of the bioactivity originally present in the crude extract is preserved during the purification process. If this is the case, it is interesting to investigate if the loss of total bioactivity is caused by the loss of material during purification or by the degradation or evaporation of potent compounds. A further possibility would be the loss of synergy between compounds present in the mixture, which disappears when the compounds are separated. In this publication, a novel formula is introduced that allows researchers to calculate total bioactivity in biological samples using experimental data from our research into the discovery of anti-inflammatory compounds from *Backhousia myrtifolia* (Grey Myrtle). The results presented show that a raw ethanolic extract retains slightly more bioactivity than the sum of all sequential extracts per gram of starting material and that—despite a large loss of material during HPLC purification—the total bioactivity in all purified fractions is retained, which is indicative of rather an additive than a synergistic principle.

## 1. Introduction

### 1.1. Therapeutic Potential of Phytochemicals in Medicinal Plants including those from Australian Rainforest Plants

Medicinal plants are an important source of active ingredients in many pharmaceutical and complementary medicine preparations [1,2,3]. As a result of geographic isolation, Australia is home to a large variety of unique and distinct flora not found elsewhere in the world. Australian plants, thriving amid the driest inhabited continent in the world, are a mostly untapped source of chemical diversity in the form of secondary metabolites [4]. Due to the harsh conditions seen in many parts of Australia, plants have developed unique survival methods and phytochemicals specific to the environmental conditions they inhabit, including intense sunlight and the change from intense rainfall to long-lasting drought. Plant-based medicines have played an important role in the health, culture, and traditions of Australian Aboriginal people, and much of our understanding of the medicinal potential of Australian native plants comes from accounts of Aboriginal ethnopharmacology [5,6].

Medicinal plants, including Australian rainforest plants, are mixtures of hundreds or thousands of bioactive compounds with potential multi-level synergistic or antagonistic interactions [7]. In the context of biology, bioactivity refers to the ability of a substance to produce a biological response or effect when applied to or introduced to a living organism. Bioactive compounds are those that are capable of interacting with living systems and influencing their function on a molecular level. These compounds can be found in a wide range of sources, including plants, animals, and microorganisms, and they can have a variety of effects on living organisms. Bioactivity can be evaluated using a variety of methods, such as cell-based assays or animal models, to determine the effects of a particular compound on living systems. The overall bioactivity of a medicinal plant extract may depend on the combined action of all these bioactive compounds, which may involve contingent, synergistic, additive, or antagonistic activity [8,9].

Bioactivity-guided purification is a method used to isolate and purify a specific compound or group of compounds from a complex mixture [10]. The process of purification *per se* is based either on increasing or decreasing the polarity of the solvent using solvent fractionation and/or chromatography. Bio-activity-guided purification involves testing the biological activity of the compound(s) in different fractions (peaks of interest) as a guide in the purification process [10]. The first step in bioactivity-guided purification is to identify the biological activity of the full-spectrum extract. This can be conducted through a variety of means, including bioassays, which are experiments that measure the biological activity, in this case, the anti-inflammatory potency. Once the biological activity of the full-spectrum extract is determined, the next step is to purify the compound(s) of interest from the complex mixture. This can be carried out through a variety of methods. One early step is the separation of the plant material by different solvents, often with increasing polarity. However, the most established and widely accepted technique is chromatography, which separates compounds based on their physical and chemical properties, and fractionation, which involves separating a mixture into smaller fractions based on the size, charge, polarity, or other properties of the compounds present. As the purification process progresses (through a reductionist approach), the biological activity of the compound(s) of interest is continuously monitored to ensure that the desired mixtures and compound(s) are being isolated and purified. Once the compound(s) of interest have been purified to the desired level, the purification process is complete, and the pure compounds can progress to structural identification [11].

At present, the chemical composition of bioactive compounds in medicinal plants is often extensively studied. However, a quantitative analysis of the contribution of each bioactive compound in a complicated mixture to the overall bioactivity as well as the loss of bioactivity in a purification process (and/or a production pipeline) is often not carried out, and our new method and formula might be useful in this context.

### 1.2. Determination of Potency as a Parameter Expression Displaying the Strength of Certain Bioactivity and the Introduction of the EDV_50_

In clinical pharmacology, the potency (specific bioactivity) of a compound is expressed as the half-maximal effective concentration (EC_50_), which refers to the concentration of a drug that induces a response halfway between the baseline and maximum [12]. While expressing the potency of a compound by its EC_50_ value makes sense in a clinical context, it is counterintuitive in the context of bioactivity-guided purification, as the potency of a compound is inversely related to its EC_50_ value, and the most potent compound is the one with the lowest EC_50_. The potency of each fraction is usually expressed as EC_50_. In detail, the term EC_50_ (or sometimes expressed as inhibitory concentration (IC_50_) for inhibitory compounds) refers to the concentration (expressed in g/L) of a drug that induces a response halfway between the baseline and maximum. The EC_50_ is the concentration (expressed in mol or g per liter) at which the compound (or extract) exhibits 50% of the maximum effect put simply, this means how many grams (for a potent drug, as few grams as possible) have to be dissolved in a liter to yield 50% of the response. In natural product chemistry, however, it would be more logical if an increase in potency were reflected by an increase in a parameter reflecting the potency. In a previous publication, the term “half-maximal effective dilution volume (EDV_50_)” as the reciprocal of the EC_50_ (1/EC_50_) was introduced, and its value increases with an increase in potency [13]. The reciprocal EC_50_ (1/EC_50_) is more like a dilution factor, describing how many liters of 1 g of extract (or compound) can be dissolved to elicit 50% of the desired activity.

For example, a herbal extract (or compound) with an IC_50_ of 1µg/mL (= 1 mg/L or 10^−3^ g/L) would have a reciprocal EC_50_ (EDV_50_) of 10^3^ L/g, meaning that 1g of the compound can be dissolved in 1000 L of water (or blood) and still lead to 50% bioactivity. It was demonstrated in a recent publication how the EDV_50_ can be used to identify potent compounds in chromatographic separations, allowing researchers to easily graph and identify anti-inflammatory compounds [13].

In this study, two examples of this approach were shown, in which an HPLC chromatogram was overlaid with the EDV_50_ to point out the most potent compounds. It was suggested that the use of the EDV_50_ will make the illustration of active fractions containing potent compounds in a chromatogram obvious to the reader and will become a useful graphic tool in natural product literature in the future [13].

In this current manuscript, the EDV_50_ will be used in a formula to allow for the calculation of the total bioactivity in a plant extract, and this bioactivity will be followed through multiple separation and purification steps of the *Backhousia myrtifolia* (Grey Myrtle) extract as a case study to enable the calculation of potential bioactivity losses.

## 2. Results

### 2.1. Introduction of the New Formula

Using already available purification yields and potency data from a previous structural identification study of anti-inflammatory compounds in Grey Myrtle [14], it was hypothesized that it is possible to quantify the overall bioactivity of a medicinal plant based on the individual potency of isolated compounds/fractions with certain bioactivity through a purification and isolation process, including sequential extraction and chromatographic separation. In most cases, for the investigation of the therapeutic potential of a medicinal plant and compound(s) of interest, the following two main questions are asked:Q1.What is the potency (EC_50_ or EDV_50_) value for the single solvent (ethanolic) extract or fraction?Q2.What is the potency (EC_50_ or EDV_50_) value for each solvent fraction (n-hexane, DCM, etc.)? This is to answer which ones are used for further purification of the most potent compounds.


However, the potency alone does not include the overall yield, and the authors wanted to ask a few more additional questions regarding the mechanism of the overall bioactivity of a medicinal plant contributed by the active compounds (e.g., a possible synergy):Q3.What is the difference in the amount of certain bioactivity between a single extraction (e.g., with EtOH) and the combined sequential extracts? Do the totals of the full spectrum and the sum of sequential extracts compare?Q4.How many units of a certain bioactivity are contained in a certain amount of a dried extract of the plant material, e.g., a raw ethanolic extract?


To do this, a helpful mathematical formula is proposed, allowing researchers to calculate total bioactivity as the key parameter to answer the two latter questions. While many natural product chemists and pharmacologists answer these questions intuitively, it is shown in this manuscript that the method and formula of total bioactivity calculation can be consistently applied.

To illustrate the equation using an example, the following section will show how to calculate the total bioactivity using the two following parameters: weight and potency.

Weight (amount of solid): Let us assume that 1 g of solids are present in the ethanolic extract, which was dissolved in 1 L (1 g/L or 1 mg/mL), serially diluted, and the bioactivity was measured for the anti-inflammatory activity of compounds.

Potency (expressed as EDV_50_): Potency can be reflected as the concentration (EC_50_) or dose (ED_50_) of a drug required to produce 50% of that drug’s effect. In a previous publication, the term “effective dilution volume (EDV_50_)” as the reciprocal of the EC_50_ (1/EC_50_) was introduced. For our formula, the potency will be expressed as EDV_50_ (1/EC_50_), as this parameter has the advantage that higher potency leads to higher EDV_50_ values. In our example, the bioactivity measured is the anti-inflammatory activity of compounds in LPS- + IFN-γ-activated macrophages using NO production as a readout. For example, let us assume that for a crude extract (e.g., in EtOH), the EC_50_ (or IC_50_) value is 1 mg/L (10^−3^ g/L). Converting it to the EDV_50_ (1/IC_50_) leads to an EDV_50_ value of 1L/mg (this value will be used in the equation below).

We introduce a novel formula as shown below to calculate the total bioactivity in an extract (or fraction or purified compound):Total bioactivity of extract X (TBA (x)) = Potency (expressed as EDV_50_) × weight (amount of the solid) of the extract

Using the values above:TBA (x) = EDV_50_ × weight, EDV_50_ = 1 L/mg, Weight: 1 g
TBA (x) = 1 L/mg × 1g = (L/10^−3^ g) × 1 g = 1000 L × g^−1^ × 1 g = 1000 L

### 2.2. Use of the Formula to Calculate Total Bioactivity in a Single Ethanolic Extract and Sequential Extracts

Sequential extraction with solvents of increasing polarity is a common method used in the isolation of natural and synthetic compounds because it allows for the separation of compounds based on their polarity. Different natural compounds have different polarities, and as a result, they will have different solubilities in different solvents. By using solvents of increasing polarity, it is possible to extract a range of compounds from a plant sample, with the less polar compounds being extracted first and the more polar compounds being extracted later in the sequence [15].

For example, nonpolar compounds such as hydrocarbons and terpenes are more soluble in nonpolar solvents such as hexane, while polar compounds such as phenols and flavonoids are more soluble in polar solvents such as methanol, ethanol, or water [16,17]. By sequentially extracting a plant sample with solvents of increasing polarity, it is possible to selectively extract different types of compounds and separate them from each other based on their polarity [15].

In the first example using the formula, the difference in total bioactivity between a single ethanolic extract and six sequential fractions was calculated. *B. myrtifolia* leaves were subjected to either a single extraction with EtOH or sequential extractions with Hexane, DCM, EtOAc, EtOH, MeOH, and water. The anti-inflammatory potency of the extracts was determined in LPS- and IFN-γ-activated RAW 264.7 macrophages using NO production as a pro-inflammatory readout (Table 1) [14].

The TBA calculation formula was used to answer the following questions:(a)How much anti-inflammatory bioactivity is contained in a certain amount of an extract?(b)For the sequential extracts, how much does one single extract contribute to the overall activity of an extract, and does the TBA of all sequential extracts equal that of the single ethanolic extract?

Single Extract: Equation (1):TBA (single ethanolic extract) = EDV_50_ × weight(1)

Sequential extract: Equation (2):
TBA (sum of sequential extracts) =TBA (hexane fraction) + TBA (DCM fraction) + TBA (EtOAc fraction) + TBA (MeOH fraction) +TBA (EtOH fraction) + TBA (water fraction)] =[EDV_50_ (hexane fraction) × weight (hexane fraction) + EDV_50_ (DCM fraction) × weight (DCM fraction), EDV_50_ (EtOAc fraction) × weight (EtOAc fraction),EDV_50_ (EtOH fraction) × weight (EtOH fraction), EDV_50_ (MeOH fraction) × weight (MeOH fraction)](2)

If TBA (single ethanolic extract) = TBA (single ethanolic extract), the TBA of all sequential extracts is equal to that of the single ethanolic extract, which may be attributed to Equation (1) the additive interaction among fractions or (2) that ethanol is capable of capturing all the active compounds that are responsible for the overall bioactivity.

If TBA (single ethanolic extract) > TBA (single ethanolic extract), the TBA of all sequential extracts exhibited a weaker effect than that of the single ethanolic extract, which may be attributed to (1) the loss of bioactivity by degradation of an active component during purification and/or (2) the loss of synergistic interaction.

If TBA (single ethanolic extract) < TBA (single ethanolic extract), the TBA of all sequential extracts exhibited a greater effect than that of the single ethanolic extract, which may be attributed to (1) the antagonistic interaction that existed in the original extract and/or the fact that ethanol is not capable of capturing all the active compounds that are responsible for the overall bioactivity.

When calculating the overall bioactivity, the TBA was highest in the sequential DCM extract (425.83 L^−1^), but this was due to the larger amount of starting material (230 g vs. 75 g). The TBA of all sequential extracts together was 856.60 L^−1^, compared to 299.40 L^−1^ for the ethanolic extract; however, this was achieved with more plant material (230 g vs. 75 g).

When the TBA was normalized to the amount of plant material, the TBA isolated per gram of dry plant was 3.99 L^−1^g^−1^ in the ethanolic extract vs. 3.72 (L^−1^g^−1^) in the combined sequential extracts (Table 2). These data indicate that the amount of total bioactivity per gram of dry plant is nearly the same for both extraction processes, suggesting that EtOH could be the solvent of choice for industrial-scale extraction. However, for further purification towards structural identification, the DCM extract should be preferred, as it will contain fewer compounds, which will simplify HPLC purification (Table 2).

### 2.3. Use of the New Formula to Calculate Total Bioactivity in Purified Compounds after HPLC Purification

In a second experimental calculation, it was determined how much of the total bioactivity from the sequential DCM extract was recovered after HPLC purification, where 17 different fractions were isolated [14]. In addition to the total bioactivity, the total yield of the sum of the fractions compared to the yield of the original TCM extract was also calculated to estimate how much material was lost during the HPLC purification process (Table 3). In an ideal world with a 100% recovery, the potency of the combined 17 fractions would be equal to that of the injected original sequential extract. The combined weight of all 17 fractions was 104.5 mg, isolated from a 4000 mg starting material of the DCM extract, yielding only a 2.61% recovery. When the TBA was calculated, a similar amount of recovery was observed (2.1%) (Table 3).

This suggests that although the loss of material during HPLC purification was immense (only 2.61% of the material was recovered), the amount of bioactivity in the purified material was mostly retained (2.1%) (Table 3). These data indicate that the individual compounds in this particular extract do not have a strong synergistic effect but rather only an additive one.

## 3. Discussion

This concept paper offers a practical approach and new formula for calculating the total bioactivity of herbal extracts, sub-fractions, and isolated compounds. This new method and formula were utilized to compare the bioactivities of sequential extracts to a raw ethanolic extract, and it was calculated that the sequential extracts contained more than 93% of the activity of the raw ethanolic extract per gram of starting material (TBA/dry plant material 3.99 vs. 3.72 (L^−1^g^−1^) (Table 2). Less than 7% of the anti-inflammatory bioactivity is lost in the sequential extraction. The loss of bioactivity could occur during the filtering, drying (e.g., by evaporation of bioactive volatile compounds), transfer of the substance into new thimbles, and repeated heating with possible degradation of active compounds.

Our formula also allowed us to choose the extract with the highest overall TBA (DCM) for further purification to have a good starting extract for HPLC purification.

In the second example, where the new formula for calculating total bioactivity was applied, the bioactivity and yields from a sequential extract (DCM) were followed through a stringent series of HPLC purifications to produce pure compounds for use in high-resolution mass spectrometry and NMR. Each substance’s potential as well as its estimated bioactivities were computed, and they were compared to the overall bioactivity found in the DCM extract (starting material). The total anti-inflammatory bioactivities of all individual compounds were computed, and it turned out that a significant loss of bioactivity was noticed during the HPLC separation. However, when the yield was considered, a similar large loss of material was observed, which could be caused by cutting the respective peaks very early to obtain pure compounds and several purification rounds. The overall TBA, however, was reduced by nearly the same amount as the total yield. This shows that the isolated, pure compounds still contained the majority of the anti-inflammatory activity. It also suggests that the compounds in this particular plant act more in an additive than a synergistic manner.

Interestingly, the second experiment also suggested that the purification and isolation of a single compound and its use as the major active compound of this extract in products would not be advised to a manufacturer of herbal medicines as the total bioactivity in our chosen plant is widely distributed over a multitude of compounds. Instead, it would be advisable to use the herbal extract or a sequential extract on its own for the herbal formulation. However, before the full extract is utilized as a source for herbal medicinal purposes, it is vital that the full chemical profiling of the plant is performed and all compounds in the extract have been identified. This is to avoid any adverse side effects that may come along with any unwanted potent (and toxic) compounds.

The authors believe it is hard to compare our findings to other studies of a comparable nature. To the best of our knowledge, no previous research on this topic has been conducted, making our approach, method, and formula unique.

Furthermore, total bioactivity calculations could be used to track the optimization of extraction and purification processes for large-scale preparations in commercial production. This could involve calculations to estimate whether bioactivity is lost during purification, for example, by oxidation and thermal degradation, or is present in the waste.

We do, however, acknowledge that our method for calculating overall bioactivity might have a significant unfamiliar novelty. The unusual and complex formula, which is not included in typical pharmacology textbooks, is one of the key drawbacks. Research students working in the chemical and pharmaceutical sciences could find it challenging to instantly understand and implement the formula. The authors are also concerned that some investigators might feel that our formal approach to adding new formulas and units would unduly complicate current research procedures. When purifying natural products to obtain pure molecules for structure identification investigations, many skilled natural product chemists instinctively calculate total bioactivity.

Our new method for determining overall bioactivity is intended to spur a range of new lines of investigation. One rationale is that it might make it possible for researchers to respond to inquiries or resolve issues that were previously challenging or impossible to handle, including tracing bioactivity from the source through the supply and production chain to the finished good. By offering fresh resources for tackling multidisciplinary issues and encouraging cooperation between other disciplines such as chemistry and pharmacology, it is hoped that the new approach will serve as an inspiration for further study.

## 4. Materials and Methods

### 4.1. Plant Material

It has to be noted that a manuscript describing the structural identification of all the compounds of *B. myrtifolia* has been published in detail and that the methods below are only an excerpt of the methods described in the published journal article [14]. The leaves of *B. myrtifolia* were collected from the Australian Botanic Garden at Mount Annan (NSW, Australia). A voucher specimen (2005-0104) has been deposited at the Australian Botanic Gardens, at Mount Annan, NSW, Australia.

### 4.2. Extraction and Bioactivity-Guided Purification of Plant Extracts

For the single ethanolic extract, fresh leaves of *B. myrtifolia* (75 g) were first cut into small pieces with scissors and then ground to a coarse powder using a hand blender, which was then extracted using absolute ethanol. The suspension was boiled for one hour, the solution was filtered using a paper filter, and it was dried in vacuo, giving a viscous, dark green, crude oily residue (3 g). For the sequential extraction, fresh leaves of *B. myrtifolia* (230 g) were crushed using a hand blender as described above and extracted sequentially using organic solvents based on their polarity (n-hexane, dichloromethane (DCM), ethyl acetate (EtOAc), ethanol (EtOH), methanol (MeOH), and finally, water) using a Buchi-811 Soxhlet Extraction system. Each extract was filtered, dried in vacuo, and weighed, and the filter residue was then used for the extraction step with the next solvent. Each extract was then subjected to anti-inflammatory testing by measuring the inhibition of NO in LPS plus IFN-γ-treated RAW 264.7 macrophages using the Griess test, and the potency was determined. For semipreparative HPLC purification, the most active sequential extract (DCM) was resuspended in EtOH and was then later subjected to semi-preparative HPLC using an Agilent C_18_ column (5 µm, 250 × 9.4 mm) column eluting at 1.8 mL/min from 10% MeCN/H_2_O to 100% MeCN (with a constant 0.01% FA modifier) over 60 min and held for a further 6 min, and then equilibrated back to 10% MeCN/H_2_O in 1 min and maintained at 10% MeCN/H_2_O for an additional 3 min, to give 17 fractions (Fr. 1–17), which included four pure fractions (Fr. 4 (6.6 mg, *t*_R_ 24.7 min), Fr.8 (4.5 mg, *t*_R_ 39.1 min), Fr.9 (4 mg, *t*_R_ 41.7 min), and Fr.15 (2.1 mg, *t*_R_ 50.3 min).

## Figures and Tables

**Table 1 ijms-24-06850-t001:** Potency and yield/weight of *Backhousia myrtifolia* extracts.

Single Extract	Potency—IC_50_ (µg/mL)	Extract Yield (g)	Starting Dry Plant Material (g)
EtOH single	10.02 ± 3.13	3.0	75
**Sequential extracts**			
Hexane	39.11 ± 6.82	1.5	230
DCM	14.09 ± 0.81	6.0	230
EtOAc	18.25 ± 7.60	2.9	230
EtOH	69.25 ± 13.33	6.9	230
MeOH	49.93 ± 8.76	3.2	230
Water	80.25 ± 17.18	5.6	230

**Table 2 ijms-24-06850-t002:** Calculations for the TBA for solvent extracts.

Column	1		2	3	4	5
	Potency IC_50_ (µg/mL)	Potency EDV_50_ (l/g)	Extract Weight (g)	TBA (L^−1^) (EDV_50_ × Weight)	Dry Plant (g)	TBA/dry Plant Material (L^−1^g^−1^)
**Single Extract**						
EtOH	10.02	99.80	3.0	299.40	75	**3.99**
**Sequential Extracts**						
Hexane	39.11	25.57	1.5	38.35	230	0.17
DCM	14.09	70.97	6.0	425.83	230	1.85
EtOAC	18.25	54.79	2.9	158.90	230	0.69
EtOH	69.25	14.44	6.9	99.64	230	0.43
MeOH	49.93	20.03	3.2	64.09	230	0.28
Water	80.25	12.46	5.6	69.78	230	0.30
All sequential extracts together			26.1	856.60		**3.72 (93.2%)**

**Table 3 ijms-24-06850-t003:** Calculations for the TBA for purified HPLC fractions 1–17.

Fraction	Weight (mg)	Potency (as IC_50_ in µg/mL)	Potency (as EDV_50_ in L/g)	TBA (L^−1^) (EDV_50_ × Weight)
F-1	1.8	68.8	14.53	0.026
F-2	9.4	57.0	17.54	0.165
F-3	9.2	62.2	16.08	0.148
F-4	6.6	44.1	22.68	0.150
F-5	6.9	35.8	27.93	0.193
F-6	7	36.0	27.78	0.194
F-7	3.5	25.6	39.06	0.137
F-8	4.5	15.9	62.89	0.283
F-9	4	16.2	61.73	0.247
F-10	6.6	16.3	61.35	0.405
F-11	10.8	17.1	58.48	0.632
F-12	12	14.5	68.97	0.828
F-13	4.9	8.6	116.28	0.570
F-14	3.5	10.3	97.09	0.340
F-15	2.1	8.3	120.48	0.253
F-16	5.7	11.5	86.96	0.496
F-17	6	6.7	149.25	0.896
Total for all 17 fractions	104.5	--	--	5.963
Original DCM extract	4000	14.09	70.97	283.88
% recovery	2.61			2.1

## Data Availability

The original data supporting the reported results are available in an original publication, which includes the structural identification of all active compounds [14].

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
