# Peer review of "A Method and Formula for the Quantitative Analysis of the Total Bioactivity of Natural Products"

_ijms, 2023, doi:10.3390/ijms24076850_

Round 1

Reviewer 1 Report (Previous Reviewer 2)

In this communication Mathew and colleagues propose a new formula to calculate total bioactivity in biological samples basing on the experimental data obtained from the experimental approach. Even though the authors indicate the limitations of this new formula, as they say it could be hard to compare to studies of comparable nature, it’s my opinion that this communication could be of interest in the scientific field of chemistry and biochemistry.

Basing on this new formula they also were able to demonstrate the efficacy of single Ethanol extraction compared to sequential extraction, opening to different applications of the formula they propose.

It’s my opinion that the article can contribute to open to new approaches and can be accepted in the present form.

Author Response

Reviewer 1:

In this communication, Mathew and colleagues propose a new formula to calculate total bioactivity in biological samples based on the experimental data obtained from the experimental approach. Even though the authors indicate the limitations of this new formula, as they say, it could be hard to compare to studies of comparable nature, it’s my opinion that this communication could be of interest in the scientific field of chemistry and biochemistry.

Based on this new formula they also were able to demonstrate the efficacy of single Ethanol extraction compared to sequential extraction, opening different applications of the formula they propose.

It’s my opinion that the article can contribute to open to new approaches and can be accepted in the present form.

Answer: Thank you for the kind comments.

Reviewer 2 Report (New Reviewer)

Title: it is recommended to remove the word novel.  

Abstract: The subject is adequately contextualized, the objective is clear, part of the methodology is described, and some results and conclusions are presented. 

Keywords: OK

Introduction: It is important to highlight one's own achievements and research, however, it is advisable to write in the third person, citing the work developed by the research group without mentioning that it is one's own.  

Results: The recurrent use of the word "novel" is not appropriate. The document should constructively highlight new findings without the use of hyperbole. 

The topic is not only interesting but also novel (it is not necessary for the authors to emphasize it repeatedly), therefore it is necessary to explain in more detail the conceptual basis of the formula, i.e., clearly the entanol extract is a crucial data, why? it can be assumed that the polarity of the extracts is an important factor. However, this reasoning may be incorrect, so the authors should explain in detail the development of the algorithm. 

Materials and Methods: The work developed by the authors is good, but it is not necessary to praise it repeatedly. The group's articles should be cited in the same way as any other article, it should only be mentioned that the methodology is described in more detail in the work [a] developed by xxxxx et al. 

Conclusions: this section cannot be changed. The novelty of the subject matter demands it. 

Author Response

Reviewer 2:

Title: it is recommended to remove the word novel.  

Answer: “Novel” was removed

Abstract: The subject is adequately contextualized, the objective is clear, part of the methodology is described, and some results and conclusions are presented. 

Answer: Thank you for the kind comments.

Keywords: OK

Answer: Thank you for the kind comments.

Introduction: It is important to highlight one's own achievements and research, however, it is advisable to write in the third person, citing the work developed by the research group without mentioning that it is one's own.  

Answer: Agreed and changed. The section was re-written, replacing “we” and “our” by writing in the third person. 

Results: The recurrent use of the word "novel" is not appropriate. The document should constructively highlight new findings without the use of hyperbole. 

Answer: “Novel” was removed

The topic is not only interesting but also novel (it is not necessary for the authors to emphasize it repeatedly), therefore it is necessary to explain in more detail the conceptual basis of the formula, i.e., clearly, the ethanol extract is crucial data, why? it can be assumed that the polarity of the extracts is an important factor. However, this reasoning may be incorrect, so the authors should explain in detail the development of the algorithm. 

Answer: This section below was added and the appropriate references were cited.

Sequential extraction with solvents of increasing polarity is a common method used in the isolation of natural compounds from plants because it allows for the separation of compounds based on their polarity. Different natural compounds have different polarities, and as a result, they will have different solubilities in different solvents. By using solvents of increasing polarity, it is possible to extract a range of compounds from a plant sample, with the less polar compounds being extracted first, and the more polar compounds being extracted later in the sequence. For example, nonpolar compounds such as hydrocarbons and terpenes are more soluble in nonpolar solvents such as hexane, while polar compounds such as phenols and flavonoids are more soluble in polar solvents such as methanol or ethanol. By sequentially extracting a plant sample with solvents of increasing polarity, it is possible to selectively extract different types of compounds and to separate them from each other based on their polarity. This method is also useful because it allows for the isolation of a wide range of compounds from a single sample, which can then be further purified and characterized using other techniques.

Materials and Methods: The work developed by the authors is good, but it is not necessary to praise it repeatedly. The group's articles should be cited in the same way as any other article, it should only be mentioned that the methodology is described in more detail in the work [a] developed by xxxxx et al. 

Conclusions: this section cannot be changed. The novelty of the subject matter demands it. 

Answer: This comment is confusing for the authors. Does the reviewer mean can or cannot (the tickbox questions indicate it should be changed)? The authors did make some changes to this section based on the other reviewer’s comments.

Reviewer 3 Report (New Reviewer)

Dear authors, 

In my opinion, the information presented in the introduction is not representative and is not sufficiently supported by sources. And most of the references are self-citations.

Author Response

Reviewer 3:

In my opinion, the information presented in the introduction is not representative and is not sufficiently supported by sources. And most of the references are self-citations.

Answer: The introduction was supported by more external references and the self-citations (including section 1.2) were removed. The introduction is now more focused on the formula.

Round 2

Reviewer 2 Report (New Reviewer)

The corresponding corrections have been made. 

This manuscript is a resubmission of an earlier submission. The following is a list of the peer review reports and author responses from that submission.

Round 1

Reviewer 1 Report

This manuscript reports the attempt to introduce a novel formula for calculating total bioactivity in biological samples. But this formal approach adding new formulas and units unduly complicates current research procedures. And there are relatively simple and reliable methods for assessing the biological properties of natural raw materials, so the proposed approach does not introduce anything new into them.

In addition, the material of the article is poorly edited, there are many repetitions. By design, the material is analytical, but some questions arise when analyzing the experimental part. So, for grinding raw materials, the authors used a manual (mechanical) blender. During such operations, local heating of the raw material occurs, which in the end can lead to incorrect results. Errors in the analysis can also be introduced by the moisture content of the feedstock. The authors report that the leaves of the plant were fresh. Since the article is analytical, the authors should have normalized this parameter. In addition, for single (ethanol) and sequential extraction with various solvents, the authors used different temperature regimes (40–100°C), which should also affect the results of the analysis. I regret but I recommend rejecting the article.

Reviewer 2 Report

In this communication Mathew and colleagues propose a new formula to calculate total bioactivity in biological samples basing on the experimental data obtained from their experimental approach. Even though the authors indicate the limitations of this new formula, as they say it could be hard to compare to studies of comparable nature, it’s my opinion that this communication could be of interest in the scientific field of chemistry and biochemistry.

Basing on this new formula they also were able to demonstrate the efficacy of single Ethanol extraction compared to sequential extraction, opening to different applications of the formula they propose.

It’s my opinion that the article can contribute to open to new approaches and can be accepted in the present form.

Reviewer 3 Report

Assessing your work as a scientific message, I must admit that it deserves to be published. It raises a controversial problem related to the method of calculating the bioactivity of natural compounds of plant origin. Your new approach to both the methodology and the calculation of the bioactivity coefficient is noteworthy. It is good that you have chosen an interesting plant (Backhousia myrtifolia; Gray myrtle) as your research subject, which has been used by the Australian Aboriginal for many years. I liked the fact that you yourself say that the new method you propose may be controversial, but that is what the development of science is all about. Considering the importance of your publication, which you unnecessarily call a communiqué, I will recommend that the Editors accept it for printing without changes.